# Effects of Antimicrobial Administration Route on Growth and Antimicrobial Resistance in Weaned Piglets

**DOI:** 10.3390/ani13203264

**Published:** 2023-10-19

**Authors:** Seong-Won Lee, Chang-Min Jung, Kyung-Hyo Do, Wan-Kyu Lee, Kwang-Won Seo

**Affiliations:** 1Boehringer Ingelheim Animal Health Korea Ltd., Yonsei Severance Bldg. 16F, 10 Tongil-ro, Jung-gu, Seoul 04527, Republic of Korea; kyssman@gmail.com; 2College of Veterinary Medicine, Chungbuk National University, Cheongju 28644, Republic of Korea; 3Onnuri Animals Hospital, Munamgil 16, Dongnam-gu, Choenan 31070, Republic of Korea

**Keywords:** swine, administration route, average daily gain, *Escherichia coli*, antimicrobial resistance

## Abstract

**Simple Summary:**

The piglets were divided into three equally sized groups (water, feed, and control). Antimicrobials were administered through drinking water and feed, in the water and feed groups, respectively, while the control group received no antimicrobial treatment. The feed conversion ratio in the water group (1.7 ± 0.78) was significantly higher than in the control (2.4 ± 1.77) and feed (2.7 ± 1.68) groups. Additionally, the route of administration did not affect antimicrobial resistance rates. Based on these results, it can be inferred that administering antimicrobials through drinking water is advantageous for pig farming.

**Abstract:**

This study aimed to determine how the route of antimicrobial administration affected the growth performance of weaned piglets. Additionally, we aimed to investigate potential differences between antimicrobial resistance developed by antimicrobials administered orally through drinking water, and those administered through feed, in weaned piglets. The research was undertaken on a farm housing 500 sows and involved 150 weaned piglets at 21 days of age. These piglets were evenly distributed into three groups of equal size: water, feed, and control. Antimicrobials were administered through drinking water and feed in the water and feed groups, respectively, while the control group received no antimicrobial treatment. The observation of piglets continued until they reached 70 days of age. The feed conversion ratio in the water group (1.7 ± 0.78) was significantly higher than in the control (2.4 ± 1.77) and feed (2.7 ± 1.68) groups. Additionally, the route of administration did not affect antimicrobial resistance rates. Based on these results, it can be inferred that administering antimicrobials through drinking water is advantageous for pig farming.

## 1. Introduction

In traditional farming setups, piglets are weaned abruptly at 3–4 weeks of age, a significantly earlier timeframe compared to the gradual transition period of 15–22 weeks observed in semi-natural rearing conditions [1]. Early weaning may lead to elevated stress levels, reduced feed intake, and poor growth performance in piglets [2]. In addition, weaned piglets are vulnerable to disease, for reasons such as declining maternal antibody titers or rapid changes in the structure and function of the small intestine [2,3,4]. Poor growth performance in piglets may cause significant economic losses for commercial swine farms [5].

Antimicrobial agents represent one of the most cost-effective methods for preserving or enhancing the health and feed efficiency of animals reared through conventional agricultural practices [6]. The pig production sector has one of the highest rates of use of antimicrobials in intensive animal production, both in terms of absolute value and treatment incidence [7]. Antimicrobials are commonly incorporated into piglet feed from birth until weaning, aiming to enhance the composition of piglet intestinal microbiota, consequently mitigating the potential repercussions of postweaning diarrhea [4]. Nevertheless, the overuse and inappropriate administration of antimicrobials in veterinary medicine have led to the emergence of antimicrobial-resistant bacteria [7,8,9]. In 2006, the European Union (EU) implemented a ban on the use of antibiotics as growth promoters [10]. This prohibition, along with the prospect of its extension to other countries, has spurred extensive research efforts focused on exploring alternative strategies that can effectively support animal health and performance [4,11]. In Korea, the use of antibiotic growth promoters in animal feeds was prohibited from July 2011 [10]. If adequate application fails, treatment may be prolonged and involve under-dosing, possibly favoring the selection of bacterial resistance [12,13].

Several studies have reported a correlation between the route of antimicrobial administration and resistance [14,15,16,17]. In pigs, oral administration is commonly employed to simultaneously deliver antimicrobials to a large population of animals [18]. Antimicrobials are administered via two main routes, both orally: via drinking water or via feed [18]. Administering antibiotics through in-water dosing is suitable for two particular scenarios in pigs: metaphylaxis and treatment. Metaphylaxis involves the preemptive treatment of animal populations that are currently dealing with varying levels of disease before the disease manifests visibly. Treatment, on the other hand, involves administering antibiotics to an individual animal or a group of animals that are already exhibiting clear clinical signs of illness [19].

The aim of a short dosing period, whether given once or at consistent intervals, is to attain both a microbiological and clinical cure [20]. During disease outbreaks among pigs, in-water antibiotic dosing is utilized for a short period until clinical signs diminish. A successful dosing event must guarantee that the majority of pigs in a group attain the required systemic exposure to the antibiotic, effectively diminishing or eradicating the specific pathogen and achieving a significant level of clinical efficacy. This approach is also designed to minimize the development and spread of antibiotic-resistant pathogens [21].

This study was conducted to assess the influence of antimicrobial administration routes on the growth performance of weaned piglets. Additionally, it aimed to investigate potential variations in antimicrobial resistance patterns when antimicrobials were administered through drinking water and feed in these piglets.

## 2. Materials and Methods

### 2.1. Experimental Design

The study was carried out on a farm housing 500 sows, with a focus on 150 weaned piglets at the age of 21 days. The experiment was approved by the Chungbuk National University Institutional Animal Care and Use Committee (CBNUA-2117-23). The piglets were segregated into three groups, each comprising 50 piglets: the water group received antimicrobials through drinking water, the feed group received antimicrobials through their feed, and the control group received no antimicrobials. Monitoring was conducted for all groups until the piglets reached 70 days of age.

### 2.2. Antimicrobial Administration

Antimicrobials were administered to piglets on the farm to mimic the preventive administration of antibiotics in pigs. Before the piglets reached 21 days of age, no antimicrobial agents were administered. The same absolute amounts of antibiotics were administered in the water and feed groups. In the water group, amoxicillin was administered once daily for 5 days for all experimental animals (100 mg/pig kg/day). In the feed group, amoxicillin was administered for 14 days for all experimental animals (100 mg/pig kg/day). The control group did not receive antimicrobials, even if they showed clinical signs of disease. If clinical signs appeared, the veterinarian used a predetermined approach, diagnosing the condition and administering either florfenicol or tylosin to the respective animals.

### 2.3. Sampling and Microbiology Analysis

Ten samples from the pigsties of each experimental group, including feces and dust, were collected at day 1 (21-day-old piglets), day 30 (50-day-old piglets), and day 50 (70-day-old piglets), to isolate *Escherichia coli*, according to the Animal and Plant Quarantine Agency guidelines [7]. A total of 30 samples from each experimental group were collected, and through this, we isolated 27 *E. coli* isolates from the water group, 29 from the feed group, and 30 from the control group. Antimicrobial susceptibility tests were performed according to the Clinical and Laboratory Standards Institute guidelines [22]. The antimicrobial disks utilized in this study were sourced from Becton-Dickinson. The experimental piglets were weighed upon entering the facility and again at 70 days of age to calculate the average daily gain (ADG) and feed conversion ratio (FCR).

### 2.4. Statistical Analysis

The statistical significance between groups was analyzed using the analysis of variance (ANOVA) and Duncan’s post hoc tests, with significance considered to be present at *p* < 0.05.

## 3. Results

We found that there were no significant differences in mortality rates between the experimental groups. Each group’s morality rate was 12.0% (6 of 50 died), 14.0% (7 of 50 died), and 16.0% (8 of 50 died) in the water group, feed group, and control group, respectively. Figure 1 shows the ADG and FCR values of the experimental groups. On day 0, the initial weights of the water group, feed group, and control group were 6.5 ± 0.73 kg, 7.9 ± 0.65 kg, and 6.9 ± 0.65 kg, respectively. After the experiment concluded (on day 50), the final weights of the water group, feed group, and control group were 21.3 ± 6.28 kg, 18.2 ± 5.77 kg, and 18.6 ± 5.21 kg, respectively. The ADG of the water group (350.4 ± 133.07 g) was significantly higher compared to that of the control (260.7 ± 113.03 g) and feed (256.4 ± 119.63 g) groups. Additionally, the FCR of the water group (1.7 ± 0.78) was significantly lower than that of the control (2.4 ± 1.77) and feed (2.7 ± 1.68) groups.

Table 1 displays the antimicrobial resistance patterns of *E. coli* isolates within each experimental group. In general, the experimental groups exhibited a comparable trend in antimicrobial resistance rates. Although there was no significant difference, amoxicillin–clavulanate resistance rates were lower in the water group (18.5%) compared with those in the feed group (24.1%). The resistance rates of cefuroxime and ceftiofur were lower in the water group (33.3% and 48.1%, respectively) compared to the feed group (37.9% and 51.7%, respectively), and the control group (40.0% and 50.0%, respectively). The water group showed lower resistance to kanamycin (74.1%), oxytetracycline (3.7%), florfenicol (55.6%), sulfisoxazole (92.6%), and trimethoprim with sulfamethoxazole (88.9%) than the feed group (82.8%, 10.3%, 62.1%, 100.0%, and 93.1%, respectively), and the control group (83.3%, 6.7%, 56.7%, 93.3%, and 93.3%, respectively).

## 4. Discussion

In this study, the objective was to investigate the impact of antimicrobial administration routes on the growth performance of weaned piglets. Additionally, potential variations in antimicrobial resistance patterns, when antimicrobials were administered through drinking water or feed in these piglets, were explored.

We found that, regarding growth performance, the ADG was significantly higher in the water group compared than that of the control and feed groups, and the FCR was significantly lower in the water group compared than that of the control and feed groups. The initial weights of the water group (6.5 ± 0.73 kg), feed group (7.9 ± 0.65 kg), and control group (6.9 ± 0.65 kg) indicate that the groups were relatively balanced in terms of weight at the beginning of the experiment. This balance is important to ensure the validity of the study. The final weights of the water group (21.3 ± 6.28 kg), feed group (18.2 ± 5.77 kg), and control group (18.6 ± 5.21 kg) show that all groups experienced weight gain over the course of the experiment. The water group had the highest final weight, suggesting that the method of antibiotic administration (water) might have contributed to better growth performance, compared to the feed and control groups. The ADG of the water group (350.4 ± 133.07 g) was significantly higher than that of the control group (260.7 ± 113.03 g) and the feed group (256.4 ± 119.63 g). This indicates that the water group exhibited faster daily weight gain, which is often considered a positive outcome in animal farming, as it implies better growth and productivity. The FCR of the water group (1.7 ± 0.78) was significantly lower than that of the control group (2.4 ± 1.77) and the feed group (2.7 ± 1.68). A lower FCR suggests that the water group required less feed to achieve the same weight gain, indicating improved feed efficiency and cost-effectiveness.

These results demonstrate that the water group had superior growth performance, as evidenced by a significantly higher ADG and a more favorable FCR, compared to the feed and control groups. This suggests that the mode of antibiotic administration, through water, may have positively influenced the overall health and growth of the experimental subjects. Further investigations could explore the specific mechanisms underlying these differences in growth and feed efficiency, which may have implications for animal husbandry practices.

Disease occurrence leads to a lower ADG and a higher FCR [2]. Diseased piglets in the feed group may reduce their feed intake, leading to suboptimal dosages of antibiotics and less efficient disease treatment. Conversely, in the water group, both diseased and healthy piglets can consume adequate amounts of water for effective disease treatment, since this approach eliminates competition for feed intake [2,5]. Thus, administering antimicrobials through drinking water facilitates efficient disease treatment, by ensuring a sufficient intake for all piglets regardless of their health status, unlike feed administration, as feed intake may be compromised in diseased individuals.

Zhang et al. reported that the antimicrobial resistance could vary depending on the route of antimicrobial administration [23]. Although there was no significant difference, we found that the resistance rates of amoxicillin–clavulanate in the water group (18.5%) were lower, when compared with those in the feed group (24.1%). Amoxicillin is effective against a wide range of bacteria [24], making it suitable for addressing a wide range of bacterial diseases in pigs [24]. The inclusion of clavulanic acid in the formulation helps combat antibiotic resistance [25]. It inhibits the action of beta-lactamase enzymes produced by some bacteria, which can otherwise render amoxicillin ineffective [26]. This preservation of amoxicillin’s effectiveness is crucial for the long-term control of bacterial infections in swine herds [26]. Amoxicillin–clavulanic acid has demonstrated a high efficacy in swine, leading to the successful resolution of many bacterial infections [24,25,26]. This can improve the overall health and well-being of pigs, leading to better growth rates and production outcomes in the swine industry.

Also, cefuroxime, the 2nd generation of cephalosporins, and ceftiofur, the 3rd generation of cephalosporins, showed a lower resistance ratio in the water group (33.8% and 48.1%, respectively) compared to the feed group (37.9% and 51.7%, respectively). Cephalosporins are commonly used in swine farms to control enteric diseases and are known for their extended duration of action, which allows for less frequent dosing [27,28,29].

The water group demonstrated a lower resistance rate to kanamycin (74.1%) compared to both the feed group (82.8%) and the control group (83.3%). This suggests that the administration of antibiotics through water may lead to a reduced development of resistance to kanamycin in the studied population. The water group showed a significantly lower resistance rate to oxytetracycline (3.7%) compared to the feed group (10.3%) and the control group (6.7%). This result suggests that the mode of antibiotic delivery, specifically through water, may play a role in lowering oxytetracycline resistance. The water group had a lower resistance rate to sulfisoxazole (92.6%) compared to both the feed group (100.0%) and the control group (93.3%). This highlights the potential benefits of using water as a delivery method for antibiotics to mitigate resistance. These results suggest that the mode of antibiotic administration, particularly through water, may influence antibiotic resistance rates in swine populations. However, the extent of this influence varies among different antibiotics, highlighting the complexity of antibiotic resistance development and the need for in-depth investigations into the underlying mechanisms. These findings underscore the importance of making informed decisions about antibiotic administration in pig farming, in order to combat antibiotic resistance effectively.

In this study, statistically significant differences in antimicrobial resistance rates were not observed, based on the route of antimicrobial administration. Equal doses of antimicrobials were given to compare the effects of different administration routes. Additionally, a duration of approximately 6–12 months is required for a significant reduction in antibiotic resistance [12,18]. Therefore, it is presumed that the observed resistance rates showed no significant differences because of the relatively short study period and equal absolute usage of antimicrobials among the groups.

## 5. Conclusions

In conclusion, we believe that administering antimicrobial agents via drinking water promotes better growth performance outcomes, as well as effective and uniform antimicrobial delivery. This approach has the potential to mitigate the development of antimicrobial resistance. These results indicate that administering antimicrobials through drinking water represents a favorable approach for disease management in pig production.

## Figures and Tables

**Figure 1 animals-13-03264-f001:**
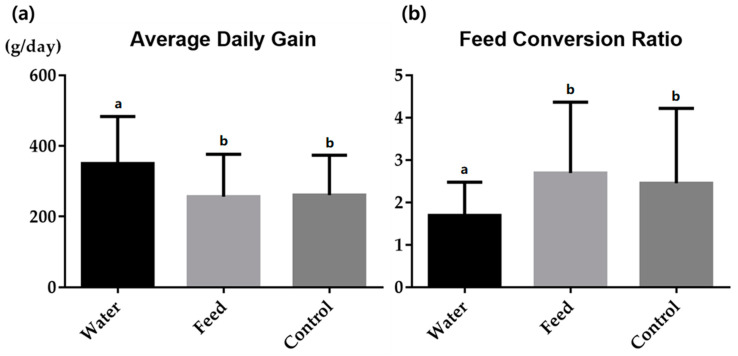
(**a**) Average daily gain and (**b**) feed conversion ratio in weaned piglets depending on the antimicrobial administration route. Water: group in which antimicrobials were administrated through drinking water; feed: group in which antimicrobials were administrated through feed; control: group which was not treated with antimicrobials. Data are expressed as means ± standard deviations. Different superscript letters (a and b) refer to statistically different groups, according to the analysis of variance (*p* < 0.05).

**Table 1 animals-13-03264-t001:** Antimicrobial resistance patterns of *Escherichia coli* isolated from 21-, 50-, and 70-day-old piglets.

Antimicrobial Class	Antimicrobial Agents (Dose/Disk)	No. of Resistant Isolates (Antimicrobial Resistance%)		
		Water (*n* = 27)	Feed (*n* = 29)	Control (*n* = 30)
Penicillins	Ampicillin (10 μg)	24 (88.9%)	25 (86.2%)	25 (83.3%)
	Amoxicillin–clavulanate (20/10 μg)	5 (18.5%)	7 (24.1%)	6 (20.0%)
	Piperacillin–tazobactam (110 μg)	23 (85.2%)	22 (75.9%)	24 (80.0%)
Carbapenems	Meropenem (10 μg)	0 (0.0%)	0 (0.0%)	0 (0.0%)
Cephalosporins	Cefazolin (30 μg)	20 (74.1%)	20 (69.0%)	23 (76.7%)
	Cefuroxime (30 μg)	9 (33.3%)	11 (37.9%)	12 (40.0%)
	Cefoxitin (30 μg)	1 (3.7%)	0 (0.0%)	0 (0.0%)
	Ceftiofur (30 μg)	13 (48.1%)	15 (51.7%)	15 (50.0%)
	Cefotaxime (30 μg)	6 (22.2%)	6 (20.7%)	10 (33.3%)
	Ceftazidime (30 μg)	0 (0.0%)	0 (0.0%)	0 (0.0%)
	Cefepime (30 μg)	0 (0.0%)	0 (0.0%)	0 (0.0%)
Aminoglycosides	Gentamicin (10 μg)	23 (85.2%)	24 (82.8%)	23 (76.7%)
	Streptomycin (10 μg)	25 (92.6%)	22 (75.9%)	25 (83.3%)
	Kanamycin (30 μg)	20 (74.1%)	24 (82.8%)	25 (83.3%)
Tetracyclines	Oxytetracycline (30 μg)	1 (3.7%)	3 (10.3%)	2 (6.7%)
	Tetracycline (30 μg)	3 (11.1%)	3 (10.3%)	2 (6.7%)
	Tigecycline (15 μg)	24 (88.9%)	28 (96.6%)	27 (90.0%)
Phenicols	Florfenicol (30 μg)	15 (55.6%)	18 (62.1%)	17 (56.7%)
	Chloramphenicol (30 μg)	23 (85.2%)	28 (96.6%)	30 (100.0%)
Quinolones	Nalidixic acid (30 μg)	15 (55.6%)	15 (51.7%)	16 (53.3%)
	Ciprofloxacin (5 μg)	8 (29.6%)	8 (27.6%)	10 (33.3%)
Sulfonamide	Sulfisoxazole (250 μg)	25 (92.6%)	29 (100.0%)	28 (93.3%)
	Trimethoprim–sulfamethoxazole (23.75/1.25 μg)	24 (88.9%)	27 (93.1%)	28 (93.3%)
Polypeptide	Colistin (10 μg)	0 (0.0%)	0 (0.0%)	0 (0.0%)

## Data Availability

The data presented in this study are available upon request from the corresponding author.

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
