# Peer review of "Effects of Antimicrobial Administration Route on Growth and Antimicrobial Resistance in Weaned Piglets"

_animals, 2023, doi:10.3390/ani13203264_

Round 1

Reviewer 1 Report

  Global comment

In this paper, the authors collected several samples from weaned piglets at 21 and 70 days of age to examine if the route of antimicrobial administration affected the growth performance of weaned piglets and if antimicrobial resistance was expressed differently when amoxicillin was administered through drinking water and feed. Isolation of E. coli was performed, as well as antimicrobial susceptibility testing. In general, the manuscript is well-written with scientific quality. However, a variety of factors can contribute to the development of antimicrobial resistance and the period of study between the first and second sampling was short.  This study allows us to study measures to mitigate antibiotic use through the route of administration and monitor the antimicrobial resistance in animals. However, I disagree with the use of antibiotics as a metaphylactic application. The methodological approach is accurate but it can be improved, therefore I suggest major revisions.

      Comments

Introduction. Nowadays, which laws or regulations exist in the Republic of Korea about antimicrobial usage in livestock as growth promotors or Metaphylaxis cases?

Line 70: change “symptoms” to clinical signs. Please, check the entire manuscript.

Line 98: add data about the concentration of each of “Antimicrobial disks” used in the text or in Table 1.

Material: How many samples were collected in each group, at 21 and 70 days of animal age? Initially, there were 50 tested animals and at 50 days of the experiment, how many animals were there? Which mortality rate?

Line 93. Please, clarify, the number of animals in each group were administrated antimicrobials different from amoxicillin? And in the control group, if some animals showed clinical signs and were administered antibiotics? Before 21 days of age, were antibiotics administrated in animals?

The results of Table 1 correspond to the phenotypic profile of E. coli isolates at 21 or at 70 days of animal age?

Author Response

Comment 1: Introduction. Nowadays, which laws or regulations exist in the Republic of Korea about antimicrobial usage in livestock as growth promotors or Metaphylaxis cases?

Response: In Korea, antimicrobial usage in livestock as growth promoters were banned in July, 2011.
According to your comment, we added line 52-53

Line 52-53:
In Korea, the use of antibiotic growth promoters in animal feeds was prohibited starting from July 201110

Comment 2: Line 70: change “symptoms” to clinical signs. Please, check the entire manuscript.

Response: According to your comment, we changed “symptoms” to “clinical signs” in entire manuscript.

Line 68, 92: Clinical symptoms ⇒ Clinical signs

Comment 3: Line 98: add data about the concentration of each of “Antimicrobial disks” used in the text or in Table 1.

Response: According to your comment, we added the doses of each of “antimicrobial disks” in Table 1.

Antimicrobial subclass

Antimicrobial agents (dose/disk)

No. of resistant isolates (Antimicrobial resistance %)

Water (n = 27)

Feed (n = 29)

Control (n = 30)

beta-lactams

Ampicillin (10 μg)

24 (88.9%)

25 (86.2%)

25 (83.3%)

Amoxicillin-clavulanate (20/10 μg)

5 (18.5%)

7 (24.1%)

6 (20.0%)

Piperacillin-tazobactam (110 μg)

23 (85.2%)

22 (75.9%)

24 (80.0%)

Meropenem (10 μg)

0 (0.0%)

0 (0.0%)

0 (0.0%)

Cephems

Cefazolin (30 μg)

20 (74.1%)

20 (69.0%)

23 (76.7%)

Cefuroxime (30 μg)

9 (33.3%)

11 (37.9%)

12 (40.0%)

Cefoxitin (30 μg)

1 (3.7%)

0 (0.0%)

0 (0.0%)

Ceftiofur (30 μg)

13 (48.1%)

15 (51.7%)

15 (50.0%)

Cefotaxime (30 μg)

6 (22.2%)

6 (20.7%)

10 (33.3%)

Ceftazidime (30 μg)

0 (0.0%)

0 (0.0%)

0 (0.0%)

Cefepime (30 μg)

0 (0.0%)

0 (0.0%)

0 (0.0%)

Aminoglycoside

Gentamicin (10 μg)

23 (85.2%)

24 (82.8%)

23 (76.7%)

Streptomycin (10 μg)

25 (92.6%)

22 (75.9%)

25 (83.3%)

Kanamycin (30 μg)

20 (74.1%)

24 (82.8%)

25 (83.3%)

Tetracyclines

Oxytetracycline (30 μg)

1 (3.7%)

3 (10.3%)

2 (6.7%)

Tetracycline (30 μg)

3 (11.1%)

3 (10.3%)

2 (6.7%)

Tigecycline (15 μg)

24 (88.9%)

28 (96.6%)

27 (90.0%)

Phenicol

Florfenicol (30 μg)

15 (55.6%)

18 (62.1%)

17 (56.7%)

Chloramphenicol (30 μg)

23 (85.2%)

28 (96.6%)

30 (100.0%)

Quinolone

Nalidixic acid (30 μg)

15 (55.6%)

15 (51.7%)

16 (53.3%)

Ciprofloxacin (5 μg)

8 (29.6%)

8 (27.6%)

10 (33.3%)

Sulfonamide

Sulfisoxazole (250 μg)

25 (92.6%)

29 (100.0%)

28 (93.3%)

Trimethoprim-sulfamethoxazole (23.75/1.25 μg)

24 (88.9%)

27 (93.1%)

28 (93.3%)

Lipopeptide

Colistin (10 μg)

0 (0.0%)

0 (0.0%)

0 (0.0%)

Comment 4: Material: How many samples were collected in each group, at 21 and 70 days of animal age? Initially, there were 50 tested animals and at 50 days of the experiment, how many animals were there? Which mortality rate?

Response: We added how many samples were collected in each group on line 95-100.
Also, we added the mortality rate on line 108-111.

Line 95-100: Ten samples from pigsties of each experimental groups, including feces and dust, were collected at day 1 (21-days-old piglets), day 30 (50-days-old piglets), and day 50 (70-days-old piglets) to isolate Escherichia coli (E. coli) according to the Animal and Plant Quarantine Agency guidelines7. Total 30 samples from each experimental groups were collected, and through this, we isolated 27 E. coli strains from the water group, 29 from the feed group, and 30 from the control group.

Line 108-111: We found that there were no significant differences in mortality rate between experimental groups. Each group of morality ratio was 12.0% (6 of 50 were dead), 14.0% (7 of 50 were dead), and 16.0% (8 of 50 were dead) in the water group, feed group, and control group, respectively.

Comment 4: Line 93. Please, clarify, the number of animals in each group were administrated antimicrobials different from amoxicillin? And in the control group, if some animals showed clinical signs and were administered antibiotics? Before 21 days of age, were antibiotics administrated in animals?

Response: Unfortunately, we didn’t explore the administered antimicrobial agents to treat clinical signs during experimental period. No antimicrobial agents were administered to control groups even if they showed clinical signs during experimental period. For reader-friendly, we revised how many animals were treated with antimicrobial agents on line 87-91.

Line 87-91: In the water group, amoxicillin was administered once daily for 5 days for all experimental animals (100 mg/pig kg/day). In the feed group, amoxicillin was administered for 2 weeks for all experimental animals (100 mg/pig kg/day). The control group did not receive antimicrobials even if they showed clinical signs.

Comment 5: The results of Table 1 correspond to the phenotypic profile of E. coli isolates at 21 or at 70 days of animal age?

Response: The results of Table 1 showed the phenotypic antimicrobial resistance profiles of E. coli isolated at 21, 50 and 70 days of animal age. .

Reviewer 2 Report

The aim of the manuscript is to evaluate the influence of the route of transmission of antimicrobials on the average daily gain, feed conversion ratio and antimicrobial resistance rate. The study examines the main routes of antimicrobial transmission in pig production.

Overall, the general idea of the study is interesting and pig production can benefit from the results, but important information is missing from the „Material and Methodes“ section as well as the „Results“ section.

General comments

Groups studied

The abstract mentions four groups studied (Line 23: drinking water, feed in the water, feed, control). Information on the feed in the water group is missing throughout the paper. Please provide the missing data or state why you excluded this group.

Data on the number of animals per group at the end of the 50-day period are missing. A mortality rate of 0% does not seem plausible, especially if the control group receives no antimicrobial at all.

Antimicrobial resistance patterns

In general, information is missing in the Material and Methodes section (Lines 95-97). How many samples were taken per matrix, age group and administration group? Are the samples taken from the same animals at two different times or at the same time but from different animals? In particular, if the samples were taken at the same time but from different animals, the conclusion in line 205 cannot be made.

The results for the samples examined are missing from the data. In how many of the samples was E. coli found? Is there a difference between faeces and dust?

In line 205 you state that you „found no increase in antimicrobial resistance rates“ but this conclusion is not supported by results Table 1 does not distinguish between different time points and does not indicate whether resistance rates have increased or decreased. In my opinion, these results also do not support the conclusions in lines 190, 194, 197 because it is unclear how the resistance rate changed over the 50-day period per group and per antimicrobial agent. The number of resistant isolates in Table 1 should be divided into the different subgroups to support the conclusions.
The discussion lacks a classification of the results with respect to the matrix tested. Please add if dust was appropriate in your context and if transfer of dust from one pigsty to another was possible and could therefore alter the results.

Specific comments

Line 47 and line 53: The statements duplicate and the references are different.

Line 89: Dosage for amoxicillin in the water group is missing.

Line 93: The control group did not recieve antimicrobials at all? Even when clinical symptoms occur?

Line 111 and Line 139: FCR is lower in the water group

Author Response

Comment 1: The abstract mentions four groups studied (Line 23: drinking water, feed in the water, feed, control). Information on the feed in the water group is missing throughout the paper. Please provide the missing data or state why you excluded this group. Data on the number of animals per group at the end of the 50-day period are missing. A mortality rate of 0% does not seem plausible, especially if the control group receives no antimicrobial at all.

Response: The experimental groups are three (antimicrobial administered through “drinking water”, “feed”, and “control group”). We revised abstract, line 23-24.

Line 23-24: Antimicrobials were administered through drinking water and feed groups respectively, while the control group received no antimicrobial treatment

Comment 2: In general, information is missing in the Material and Methodes section (Lines 95-97). How many samples were taken per matrix, age group and administration group? Are the samples taken from the same animals at two different times or at the same time but from different animals? In particular, if the samples were taken at the same time but from different animals, the conclusion in line 205 cannot be made. The results for the samples examined are missing from the data. In how many of the samples was E. coli found? Is there a difference between faeces and dust?

Response: Each of 10 samples from each experimental group are collected at day 1, 30, and 50, respectively. Also, during the experimental period, 27, 29, and 30 E. coli strains were isolated from water, feed, and control group, respectively. According to your comment, we revised Materials & Methods on line 95-100. There was no considerable difference on E. coli isolation rates between feces and dust samples. Forty-four E. coli strains were isolated from total 45 fecal samples, and 42 strains were isolated from 45 dust samples. 

Line 95-100: Ten samples from pigsties of each experimental groups, including feces and dust, were collected at day 1 (21-days-old piglets), day 30 (50-days-old piglets), and day 50 (70-days-old piglets) to isolate Escherichia coli (E. coli) according to the Animal and Plant Quarantine Agency guidelines7. Total 30 samples from each experimental group were collected, and through this, we isolated 27 E. coli strains from the water group, 29 from the feed group, and 30 from the control group

Comment 3: In line 205 you state that you „found no increase in antimicrobial resistance rates“ but this conclusion is not supported by results Table 1 does not distinguish between different time points and does not indicate whether resistance rates have increased or decreased. In my opinion, these results also do not support the conclusions in lines 190, 194, 197 because it is unclear how the resistance rate changed over the 50-day period per group and per antimicrobial agent. The number of resistant isolates in Table 1 should be divided into the different subgroups to support the conclusions. The discussion lacks a classification of the results with respect to the matrix tested. Please add if dust was appropriate in your context and if transfer of dust from one pigsty to another was possible and could therefore alter the results.

Response: Thank you for  this suggestion. It would have been interesting to explore this aspect. However, in the case of our study, if the data for Table 1 were divided, there were too low samples so, statistical analysis are not possible. So we believe that Table 1 should not be divided.

Comment 4: Line 47 and line 53: The statements duplicate and the references are different.

Response: According to your comment, we revised the reference for line 47-49, and deleted line 53.

Line 47-49: Nevertheless, the overuse and inappropriate administration of antimicrobials in veterinary medicine have led to the emergence of antimicrobial-resistant bacteria7, 8, 9.

(Deleted) Line 53: However, antimicrobial-resistant bacteria have emerged due to excessive and inappropriate use of antimicrobials in veterinary medicine.

Comment 5: Line 89: Dosage for amoxicillin in the water group is missing.

Response: According to your comment, we added the dosage for amoxicillin in the water group on line 89.

Line 87-89: The same absolute amounts of antibiotics were administered in the water and feed groups. In the water group, amoxicillin was administered once daily for 5 days for all experimental animals (100 mg/pig kg/day).

Comment 6: Line 93: The control group did not recieve antimicrobials at all? Even when clinical symptoms occur?

Response: No antimicrobial agents were administered to control groups even if they showed clinical signs during experimental period. According to your comment, we revised line 90-91.

Line 90-91: The control group did not receive antimicrobials even if they showed clinical signs.

Comment 6: Line 93: The control group did not recieve antimicrobials at all? Even when clinical symptoms occur?

Response: No antimicrobial agents were administered to control groups even if they showed clinical signs during experimental period. According to your comment, we revised line 90-91.

Line 90-91: The control group did not receive antimicrobials even if they showed clinical signs.

Comment 7: Line 111 and Line 139: FCR is lower in the water group

Response: According to your comment, we revised line and line 116-118, and deleted FCR on line 139.

Line 116-118: Additionally, FCR in the water group (1.7 ± 0.78) was significantly lower than that in the control (2.4 ± 1.77) and feed (2.7 ± 1.68) groups.

(Deleted) Line 139: We found that the growth performance; ADG and FCR was significantly higher in the water group compared than that in the control and feed groups.

Round 2

Reviewer 1 Report

Many thanks to the authors for their response. 

Before the manuscript publication, I propose to to mention in title of Table 1 that these results showed the total phenotypic antimicrobial resistance profiles of E. coli isolated at 21, 50 and 70 days of animal age. To improve the results, in this table can be also included the results obtained at 21, 50 and 70 days separately. 

Author Response

According to your comment, we revised the title of Table 1.

Before: Table 1. Antimicrobial resistance patterns of Escherichia coli isolates

After: Table 1. Antimicrobial resistance patterns of Escherichia coli isolated from 21, 50, and 70 days of experimental piglets.

Also, thank you for suggestion to improve the results of Table 1.

However, in the case of our study, if the data for Table 1 were divided, there were too low samples so, statistical analysis are not possible.

We think if we scribe the 21, 50, and 70 days seperately in Table 1, it could confuse the reader, so  so we believe that Table 1 should not be divided.

Reviewer 2 Report

Line 144-145: Please revise the sentence. Some part seems to be missing.

Line 210-211: "In this study, we found no increase in antimicrobial resistance rates depending on the route of antimicrobial administration."
As already mentioned, in my opinion any change in rates is not supported by the results. However, it would be possible to say that there are "no statistically significant differences in ...."

Everything else is fine for me.

Author Response

As your comment, we revised line 144-146, and line 211-212.

Line 144-146: We found that the growth performance; ADG was significantly higher in the water group compared than that in the control and feed groups, and FCR was significantly lower in the water group compared than that in the control and feed groups. 

Line 211-212: In this study, we found no statistically significant differences in antimicrobial resistance rates depending on the route of antimicrobial administration.